# Characteristic Binding Landscape of Estrogen Receptor-α36 Protein Enhances Promising Cancer Drug Design

**DOI:** 10.3390/biom13121798

**Published:** 2023-12-14

**Authors:** Adeniyi T. Adewumi, Salerwe Mosebi

**Affiliations:** Department of Life and Consumer Sciences, University of South Africa, Private Bag X06, Florida 1710, South Africa; adewuat@unisa.ac.za

**Keywords:** Breast cancer, Tumour Threat, Estrogen receptor-α36, Structure-based inhibitors, Broussoflavonol B, All-atoms MD simulations, Binding mechanism

## Abstract

Breast cancer (BC) remains the most common cancer among women worldwide, and estrogen receptor-α expression is a critical diagnostic factor for BC. Estrogen receptor (ER-α36) is a dominant-negative effector of ER-α66-mediated estrogen-responsive gene pathways. ER-α36 is a novel target that mediates the non-genomic estrogen signaling pathway. However, the crystallized structure of ER-α36 remains unavailable for molecular studies. ER-positive and triple-negative BC tumors aggressively resist the FDA-approved drugs; therefore, highly potent structure-based inhibitors with preeminent benefits over toxicity will preferably replace the current BC treatment. Broussoflanol B (BFB), a *B. papyrifera* bark compound, exhibits potent growth inhibitory activity in ER-negative BC cells by inducing cell cycle arrest. For the first time, we unravel the comparative dynamic events of the enzymes’ structures and the binding mechanisms of BFB when bound to the ER-α36 and ER-α66 ligand-binding domain using an all-atom molecular dynamics simulations approach and MM/PBSA-binding-free energy calculations. The dynamic findings have revealed that ER-α36 and ER-α66 LBD undergo timescale “coiling”, opening and closing conformations favoring the high-affinity BFB-bound ER-α36 (ΔG = −52.57 kcal/mol) compared to the BFB-bound ER-α66 (ΔG = −42.41 kcal/mol). Moreover, the unbound (1.260 Å) and bound ER-α36 (1.182 Å) exhibit the highest flexibilities and atomistic motions relative to the ER-α66 systems. The RMSF (Å) of the unbound ER-α36 and ER-α66 exhibit lesser stabilities than the BFB-bound systems, resulting in higher structural flexibilities and atomistic motions than the bound variants. These findings present a model that describes the mechanisms by which the BFB compound induces downregulation-accompanied cell cycle arrest at the Gap_0_ and Gap_1_ phases.

## 1. Introduction

The death risk of breast cancer (BC) (14%) and prostate cancer (PC) (9%) has remained significantly high, and these cancers are the most prevalent types of cancer among women and men, especially in the Western world [1]. Generally, the disease can be managed by several therapy strategies, including chemotherapy, radiotherapy, targeted therapy, surgery, combined therapy (CT), prevention, and endocrine therapy (ET) [2]. ET is an inflexion point, including selective estrogen receptor modulators (SERMs), selective estrogen receptor downregulators (SERDs), and aromatase inhibitors (AI) [3,4]. Estrogen therapy is one of the most necessary treatments for estrogen receptor (ER)-positive BC. Estrogen signaling is critical to breast cancer initiation and development. However, abnormal levels of human estrogen and androgen hormones cause carcinogenesis of the breast and prostate, respectively [1,5]. About 70% of all types of tumor expression are transcription factors that activate estrogen binding and regulate various gene expressions during proliferation [3]. Human estrogen receptor alpha (hERα or ER-α66) is a variant of ER that is predominantly expressed in BC [3] compared to the Erβ isoform; therefore, it is a critical target in the assessment of prognosis [6] and the design of a BC therapy strategy. Furthermore, ER-α66 cDNA cloning produces a unique spliced ER-α36 protein of 36 kDa, which lacks the transcriptional activation domains (AF-1 and AF-2) of ER-α66 [7]. Nevertheless, ER-α36 possesses conserved DNA-binding (DBD), ligand-binding (LBD), and dimerization domains [7]. In addition, ER-α36 has three myristoylation sites proximal to its N-terminal [6]. Moreover, it is functionally different from ERα-66 and highly associated with the plasma membrane, acting as a negative effector of estrogen-dependent and -independent transactivation of ER-α66 and ER-β [6,8]. ER-α36 is a dominant-negative effector of ER-α66-mediated estrogen-responsive gene pathways and mediates the non-genomic estrogen signaling pathway through EGFR/Src/ERK [9].

Estrogen receptor alpha is characterized by acquired mutations in hormone-receptor-positive (HR+), negative, and metastatic breast cancer (MBC), which makes anticancer drug design challenging [3]. The treatment of endocrine resistance is complicated, causing ET, and even CT, to be ineffective [10]. About 40% of diagnosed HR+ MBC patients treated with ET relapse over time, while treated prostate cancer patients show inconsistencies in ER expression and efficacy of SERMs [11]. ER-α36 dysregulation is hugely associated with various diseases. ER-α36 correlates with a larger tumor size and more distant metastasis and is associated with advanced disease severity and the poor survival of breast cancer patients [12]. Moreover, ER-α36 acquires resistance to ET drugs by activating non-genomic signaling pathways, which provides insights into why resistance characterizes some breast cancers while others are not improved when using an antiestrogen strategy.

However, the molecular mechanisms underlying tamoxifen resistance to inhibiting the function of ER-α36 are still largely unclear. Another problem is that the impact of ER-α36 mutation on the immune modulatory effect of estrogen and the ER-α36 inhibitors is unknown. Computer-based drug design techniques provide a platform that covers the shortcomings of various wet laboratory-based drug design techniques and methods. Interestingly, proteins’ geometry and topology parameters, such as cross channels, interior cavities, and surface pockets, are fundamentally critical for the proteins to play their roles. The impacts of dynamics and motions on the structure and enzymatic function of modeled ER-α36 LBD (ligand-binding domain) upon binding fulvestrant (reference inhibitor) and broussoflavonol B (BFB) were investigated using integrated all-atom molecular dynamics simulations. Fulvestrant is an FDA-approved SERD used to treat post-menopause in hormone-receptor-positive advanced (or metastatic) BC, which is also active in patients treated with therapies other than ET [13].

The ICI 182,780 (fulvestrant, Faslodex), a ‘pure’ antiestrogen, accelerates the degradation and impairs the dimerization and nuclear localization of ER-α66. However, several studies have reported that ICI 182,780 failed to induce the degradation of ER-α36, presumably because ER-α36 has a truncated ligand-binding domain, lacking the last four helices (helix 9–12) of ER-α66, which are essential for protein degradation induced by ICI 182,780. Therefore, the failure of ER-α36 degradation is a highly possible reason for ICI 182,780 resistance.

However, studies have reported processes that drive acquired resistance to fulvestrant therapy [14]. Recent works suggest that ESR1 and PIK3CA mutations and ER/HER2 crosstalk underly the molecular mechanisms for fulvestrant resistance [15,16]. Broussonetia papyrifera (paper mulberry) is a plant that grows naturally in Asia and Pacific countries that contains numerous flavonoid compounds. Among the compounds, broussoflavonol B (BFB; 5,7,3′,4′-tetrahydroxy-3-methoxy-6,8-diprenylflavone) exhibits various activity types against targets. Such protein targets include tyrosinase, aromatase, secretory phospholipase A-2, nitric oxide synthase, and ER-positive breast cancer [17,18]. Broussoflavonol B induces cell cycle arrest at the G_0_/G_1_ and G_2_/M phases via downregulation of c-Myc protein [18]. The compound induces apoptotic cell death, accumulating the annexin-V- and propidium-iodide-positive cells and the cleavage of caspases [17]. The report further showed that the BFB-based treatment decreased the steady-state levels of ER-α36, restricting the growth of the stem-like cells in ER-negative breast cancer MDA-MB-231 [17]. The studies reported the IC_50_ values of broussoflavonol B, which are assay-specific, and that validation with computational results is unreliable [19]. Moreover, the inhibition dynamics of BFB binding to any target have not been studied, especially ER-α36 and ER-α66. Therefore, in this study, we build a 3D structure (homology model) of ER-α36 LBD (ligand-binding domain) in order to gain insights into its dynamics and the motions of the ER-α36 structure and function compared to ER-α66, hypothesizing their identically conserved LBD. The research gap relating to the dynamics of the mechanism, structure, and function of ER-α36 LBD prompted the undertaking of this project, which could aid in designing new efficacious drugs in the context of the human estrogen receptor ER-α36 LBD.

## 2. Methodology

### 2.1. Homology Modeling

ER-α36 is the primary study target, and its 3D conformational structure was unavailable. Therefore, the primary structure (amino acid sequence) of ER-α36 was retrieved from UniProt.org [20] and uploaded onto the SwissModel BLAST database (SMD), automatically searching for possible structural templates (Accession ID: P03372-4), which eventually created a homology model. The ER-α36 auto-model criteria by SMD were primarily based on significant correlation with the target (ER-α36) protein, maximum sequence identity (>30%), good sequence similarity, query coverage, and acceptable crystallographic data. The global model quality estimate (GMQE) and quaternary structure quality estimate (QSQE) were also used to evaluate the quality of the model.

Additional validation methods were used to model ER-α36, including the modeler-10.3 add-on UCSF Chimera tool, ConSurf, and AlphaFOLD (FunFOLD) servers [21]. The three (3) selected templates used to model on the UCSF Chimera-Modeler included 1R5K.pdb [22], 2P15.pdb [23], and 6DF6.pdb [24], which were human X-ray-crystallized ER-α66 LBD proteins. A multiple sequence alignment was performed using the EMBL-EBI ClustalW server [25]. ClustalW uses a BLOSUM matrix of 10 for gap opening and a penalties gap of 0.1 extensions for gap extension [26]. The SMD model was superimposed on the models obtained from the other three methods to validate the protein. Further validation was performed by running a 40 ns molecular dynamics simulation for ER-α36 LBD. A plot of Ramachandran bond angles and torsional strain analysis was obtained using Maestro-13.0 software (http://www.schrodinger.com/; 14 October 2023)

### 2.2. Active Site Identification

Accurately predicting ligand-binding residues from amino acid sequences is crucial for the automated functional annotation of novel proteins. The possible binding sites of ER-α36 LBD were predicted using web servers, including the Computer Atlas of Surface Topography of proteins (CASTp 3.0) [27] and the FunFOLD method incorporated in the IntFOLD integrated protein structure and function prediction server (version 6.0). CASTp 3.0 provides online services for locating, delineating, and measuring protein structures’ geometric and topological properties. The CASTp 3.0 server identifies all surface pockets, interior cavities, and cross channels in a protein structure. The method uses the alpha shape method [28] of computational geometry to identify the topographic features and estimate the available entrance openings’ volumes, areas, and sizes. The volume, areas, and sizes are analytically computed using Cannolly’s molecular surface model [29] and Richards’ solvent-accessible surface model [29]. It calculates the secondary structures of proteins using DSSP [30] by obtaining the residue annotations of proteins from the UniProt database and mapping them to the PDB database. FunFOLD is benchmarked against the top servers and the manual prediction groups tested at CASP8 and CASP9 [21]. The FunFOLD method uses an intuitive approach for cluster identification and residue selection to predict the ligand-binding site of a protein. The IntFOLD-incorporated FunFOLD server integrates into existing fold recognition servers, requiring only a 3D model and lists of templates as inputs.

### 2.3. Molecular Docking Calculations

The docking calculations were performed for broussoflavonol B and fulvestrant (reference inhibitor) (Figure 1). The molecular structure of BFB was downloaded from PubChem, while the structure of fulvestrant (FULV) was drawn in MarvinSketch (http://www.chemaxon.com, accessed on 28 November 2023) and converted to mol2 format. Broussoflavonol B and FULV have been individually assessed in Molegro Molecular Viewer (MMV) software (http://www.molegro.com/, accessed on 28 November 2023) to ensure suitably displayed bond angles and a hybridization state. The inhibitors were minimized and optimized using Avogadro’s steepest descent method and the GAFF force field [31]. The AutoDock tool’s graphical user interface was used to generate the grid dimension and center of the box for the final chosen binding pocket. The defined ER-α36 grid box (Å) parameters are center (*X* = 6.28, *Y* = −0.55, *Z* = −6.32) and dimensions (*X* = 22.39, *Y* = 24.40, *Z* = 19.66). In the same vein, the grid box at the active site of the ER-α66 protein was defined as center (*X* = 13.14, *Y* = 10.73, *Z* = −21.93) and dimensions (*X* = 25.50, *Y* = 22.18, *Z* = 27.14). The docking calculations were performed with AutoDock Vina using the Lamarckian genetic algorithm [32]. The protein and ligands were prepared using UCSF Chimera software-1.16 by adding hydrogen and Gasteiger charges and assigning the atom types with the AutoDock tool before the calculations were performed.

### 2.4. Molecular Dynamics (MD) Simulations [33]

All-atom molecular dynamics simulations were carried out for the ligand-bound BFB-bound ER-α36, BFB-bound ER-α66, and FULV-bound ER-α36 complexes using the best-fit poses docked at their selected active sites. The MD simulations were performed in the AMBER18GPU version of the PMEMD-CUDA engine in the AMBER18 package in the Lengau CHPC server (http://www.ambermd.org, accessed on 28 November 2023) [34]. The hydrogen atoms and Gasteiger charges were added using the UCSF Chimera tool (http://www.cgl.ucsf.edu/chimera, accessed on 28 November 2023). The partial atomic charges for inhibitors BFB and FULV were calculated using the restrained electrostatic potential (RESP), and the ANTECHAMBER module within AMBER was used to generate the force field parameter using the general AMBER force field (GAFF) partial charges [31]. The GAFF assigned atom types and filled in the missing parameters. The molecules and proteins parametrized were the AMBER FF14 force field version [31]. The AMBER LEAP module was utilized to add the protons and the required Na+ ions (counterions) for protein neutralization. Furthermore, the system was explicitly solvated with a box of TIP3P water molecules at a distance of 12.0 Å from the protein atoms [35]. The MD simulations were run using the periodic boundary conditions, while obtaining long-range contributions from the electrostatics from the particle mesh Ewald (PME) method contained in AMBER 18, with direct space and van der Waals interactions restricted to 12.0 Å. The unbound systems and complexes were initially minimized for 2500 steps with 500 kcal/mol Å^2^ restraint potential, and a whole minimization step of 5000 steps was further run without restraint using the conjugate algorithm. The gradual heating was carried out in a canonical ensemble (NVT) (between 0 and 300 K), executed for 5 ps, fixing several atoms and volumes. A potential harmonic restraint of 10 kcal/mol Å^2^ was used to restrain the solute systems with a collision frequency of 1.0 ps^−1^. The systems were equilibrated for 1 ns while keeping the constant operating temperature of 300° K without restraints. All of the hydrogen bonding types were constrained using the SHAKE algorithm. The number of atoms and the pressure were kept constant using an isobaric isothermal ensemble (NPT), Barendsen barostat was used to hold constant pressure at 1 bar, and the MD simulation was carried out over 200 ns [36,37]. The coordinates and trajectories were printed and analyzed every 1ps using the PTRAJ module in AMBER18GPU. The structural conformations of unliganded and complex systems were investigated statistically for stability and flexibility by estimating the root mean square deviations (RMSD) [38] and the root mean square fluctuations (RMSF) [37], respectively. In addition, the atomic displacement of the enzyme was investigated using principal component analysis (PCA). The details of the methods, including the PCA [39], have been reported in our previous works several times. The data were processed with MicroCAL Origin 6.0 software [40], while UCSF Chimera and Discovery Studio 2021 [41] were used for visualization and structural analysis, such as for the ligand-binding mechanism.

### 2.5. Thermodynamic Parameter Calculations

The binding affinities of the bound and unbound systems were obtained by computing the binding-free energy using the molecular mechanics/GB surface area method (MM/GBSA) [42]. The free energy was calculated based on the average of 10,000 snapshots from a 40 ns trajectory. The estimated binding free energy, ΔG, for each molecular system, including the complex, inhibitor, and protein, can be given as follows [43]:ΔG_bind_ = G_complex_ − G_protein_ − G_inhibitor_(1)
ΔG_bind_ = E_gas_ + G_sol_ − TS(2)
E_gas_ = E_int_ + E_vdw_ + E_ele_(3)
G_sol_ = G_GB_ + G_SA_(4)
G_SA_ = γSASA(5)

Moreover, the details of the method can be found in our previous work [38].

## 3. Results and Discussion

### 3.1. Structure of the ER-α36 Model and Its Binding Pockets

Some in vitro and in vivo assay findings have been reported on the activity and inhibition of ER-α36, emphasizing the ligand-binding domain (LBD) as a binding pocket. This protein has a significant conserved amino acid sequence with the ER-α66 variant. The amino acids within the LBD include those between 302 and 430. Moreover, the DNA-binding and 27 amino acid domains, whose roles are still unclear, have been reported. Preparing the 3D crystal structure of protein and ligands precedes docking calculations or other steps in ligand-based drug design. Neither the unbound 3D crystal structure of the ER-α36 nor the ER-α36 co-crystallized structure with broussoflavonol B (study inhibitor) was available during this study. Therefore, a homology model was built. The BLASTs of the protein data bank (PDB) database search revealed the 3D templates that we used to build the validated model. A total of 50 templates were used in the SWISS-MODEL algorithm based on the criteria stated in Section 2.1 to build 15 models. The best two models based on satisfactory criteria were selected and superimposed on the Chimera tool for conserved residue comparison of the amino acid sequences. Table 1 details the templates that were used to obtain the best two SWISS-MODELs.

Interestingly, the SWISS-MODEL and the ConSurf servers predicted 3D structures without the DNA-binding domain (DBD) and Hinge domain (128 amino acid sequences). In contrast, the AlphaFOLD server and Chimera-Modeler 10.3 add-on model predictions included DBD and Hinge region amino acid sequences. However, deleting the DBD and Hinge domain of the AlphaFOLD server and the Chimera-Modeler models [19] produced the exact same ER-α36 LBD 3D homology model seen in the SWISS-MODEL and ConSurf models.

Regarding the Chimera-Modeler 10.3 methods, three templates were selected to reduce the structural or amino acid differences encountered, using only one template. The regions with the missing residues in the templates were confirmed to be regions outside of the critical active site. A zDOPE value >0.7 reflects ≥95% probability of the model with the correct fold, indicating a reliable model structure, while a negative value indicates a better model. Out of the three models generated on the Chimera-Modeler interface in this study, the best mode gave a zDOPE value of 0.41. A positive zDOPE model might have occurred due to the amino acid sequences of the template. The ConSurf, AlphaFOLD, and UCSF Chimera-Modeler interface models were superimposed with the SWISS-MODEL and observed on the UCSF Chimera tool to look for a significant correlation (Figure 2).

A Ramachandran plot of SWISS-MODEL ER-α36 (Figure 3) was generated in Maestro in order to evaluate the stereochemistry of the ER-α36 structure model. The plot revealed that 97.10% of the residues are located within the most favored region, while 1.74% and 1.16% are located in the additionally and generously allowed regions, respectively. The angle metrics of the ER-α36 residues GLY193 (*phi* (*φ*) = 93.5° *psi* (*ψ*) = −12.3°), GLY227 (*φ* = 81.6°, *ψ* = −6.1°), and GLY247 (*phi* = 78.2° and *psi* = −3.5°) are found within the generously allowed region. As justified, the angle distribution is good, with more than 95% of the residues occurring within the favored regions, showing suitably modeled ER-α36 geometry and stereochemistry distributions.

The 3D structures of ER-α36 obtained from the four methods (SWISS-MODEL, Modeler, ConSurf server, and AlphaFOLD) were manually inspected in order to compare their correlation with the 3D structures model from the Psipred outcome (Figure 4). Interestingly, the homology models showed an excellent correlation with the Psipred result.

Excluding the DNA and Hinge domains, the 3D secondary structures of the modeled ER-α36 protein consist of two beta-strands with corresponding residues in the following brackets: β_1_-strand (residues: 235–238) and β_2_-strand (residues: 227–232). The model has the following eight helices, with the corresponding residues in brackets: α_1_-helix (139–149), α_2_-helix (166–190), α_3_-helix (194–196), α_4_-helix (199–221), α_5_-helix (239–244), α_6_-helix (248–265), α_7_-helix (269–282), and α_8_-helix (293–308). Compared to the ER-α66 variant, the ER-α36 LBD lacks helices 9–12; therefore, the ER-α36 protein LBD conformation differs from the ER-α66 [44].

The homology models obtained for the SWISS-MODEL and ConSurf servers did not contain 1–118 residues (residues within the DNA and Hinge domains) [45], yielding an incomplete ER-α36 ligand-binding domain. On the other hand, the AlphaFOLD and Chimera-Modeler models produced full-length ER-α36 protein domains. The ligand-binding domain (LBD) of ER-α36 was fully conserved with ER-α66 LBD, but the truncating of ER-α36 LBD may impact its overall binding of ligands.

Although the pocket residues of ER-α66 have been investigated in order to dock a ligand to a model of ER-α36 LBD based on sequence conservation, the truncated region of the ER-α36 protein confirmed that it has a different conformation. Therefore, the Castp 3.0 server and IntFOLD were used to identify the possible active residues within the ER-α36 LBD protein in order to validate the binding pocket. Both of the active site identification programs predicted almost 100% identical binding residues (Figure 5). The agreement between the Castp and IntFOLD results confirms the locality of the ER-α36 LBD binding pocket and its conserved identity with the ER-α66 LBD. Moreover, the predicted pocket agrees with the reported binding residues of the ER-α36 homology model produced by Kang et al. based on 1ERE.pbd and the templates (ER-α66 LBD) in Table 1, including 7PRW, 1R5K, 2P15, and 6DF6. Figure 6 shows that the amino acids in the ER-α36 LBD pocket are entirely identical to those of the ER-α66 LBD pocket.

### 3.2. Docking Affinity of the Ligands Bound to ER-α36 and ER-α66

Broussoflavonol B and fulvestrant were used in the docking experiments to explore the comparative binding landscape of ER-α36 and ER-α66, even though this study is more critical about the ER-α36 variant. The results have revealed that the poses for both BFB and FULV comfortably fit into the ER-α36 truncated LBD compared to BFB docked to the full-length ER-α66 LBD.

Table 2 shows the binding affinity of broussoflavonol and fulvestrant to their respective ER-α66 and ER-α36 proteins. The results indicate that broussoflavonol had a higher binding affinity (−7.7 kcal/mol) with ER-α36 than ER-α66 and FULV-bound ER-α36. Similarly, it had a higher binding affinity than fulvestrant, with a −7.3 Kcal/mol affinity. The binding affinity values provide insight into the strength of the interaction between the ligands and their respective target molecules, which can contribute to their overall efficacy.

In the BFB-bound ER-α36 complex, the ligand forms strong polar H-bonds with the side chain of GLU180 and forms hydrophobic interactions, including pi interactions with PHE231, alkyl hydrophobic interactions with ALA177 and LEU211, and several mixed pi/alkyl hydrophobic interactions with TRP210, LEU214, and PHE231. In comparison, the polar residues within the binding pocket of ER-α36 did not show contact with FULV. On the other hand, the BFB-bound complex shows no contact with the ER-α66 LBD polar pocket. These docking data agree with the experimental IC_50_ and reveal the trend in the downregulation of ER-α36 expression inhibition of proliferation in ER-positive breast cancer or growth-inhibitory activity in ER-negative breast cancer [18].

### 3.3. Molecular Dynamics Simulation Analysis

#### 3.3.1. Thermodynamics Calculations Using MMGB(PB)SA Methods 

The experimental IC_50_ data from the in vitro and in vivo study of ER-α36 and the inhibitory activity of the BFB have been reported in various studies, and the binding-free energies of the complex system have never been investigated. In contrast, Kollman et al., 2000, stated that the experimental binding-free energies are more accurate and reliable than the IC_50_ values. Therefore, we computed the binding-free energy using MMGB(PB) methods and compared the correlations with the docking results (Table 2). There was a significant correlation between the MMGB(PB)SA (Table 3) and the docking calculations when comparing the ER-α36 LBD to the ER-α66 LBD for BFB binding. The computed binding-free energies were significantly higher at the BFB-bound ER-α36 LBD site (−52.57 kcal/mol) than those found at the ER-α66 LBD site for BFB (−42.41 kcal/mol). However, the ΔG_bind_ of the ER-α66 LBD for BFB was slightly higher than that found in the ER-α36 LBD for FULV (−37.43 kcal/mol), contrary to the docking results; nevertheless, the difference was much lower when we compared the ΔG_bind_ of the BFB-bound ER-α36 and the BFB-bound ER-α66. Therefore, we conclude that the ER-α36 ligand-binding domain is a novel target, and these interaction mechanisms can be explored to redesign the anticancer FDA-approved drugs facing resistance or mutations or to design newer, promising inhibitors in the clinical stages of drug development.

#### 3.3.2. Contrasting the Interaction Channels of the ER-α Variants Systems

The 3D interaction channels (Figure 7B, Figure 8B and Figure 9B) indicate the predicted interactions of BFB and FULV with the active sites of the ER-α36 and the interaction networks of BFB with the ER-α66 proteins. The novel ER-α36 truncated binding residues gave significantly more interaction options with BFB than FULV, even compared to the number of interactions between the ER-α66′s full-length active site and BFB.

BFB and FULV interact with ER-α36′s truncated hydrophobic active site, as follows: MET170, LEU173, GLU180, LEU211, MET215, and LEU218. At the binding site of the ER-α36 protein, BFB interacts with ASP178 and LEU255. The following ER-α36 active site residues lie within the ER-α36 novel truncated binding interactions (with their ER-α66 binding pocket equivalent residues enclosed in brackets): MET170 (343), LEU173 (346), THR174 (347), LEU176 (349), ALA177 (350), GLU180 (353), TRP210 (383), LEU211 (384), LEU214 (387), MET215 (388), LEU218 (391), ARG221 (394), PHE231 (404), VAL245 (418), MET248 (421), ILE251 (424), PHE252 (425), and LEU255 (428). The energy contributions in kcal/mol by the ER-α36 pocket residues in binding broussoflavonol B are as follows: MET170 (−1.2), LEU173 (−2.5), ALA177 (−1.63), GLU180 (−7.6), TRP210 (−0.5), LEU211 (−1.12), LEU214 (−1.5), MET215 (−1.05), LEU218 (−1.63), PHE231 (−1.19), VAL245 (−0.53), and MET248 (−1.71). GLU180 contributes significantly higher energy binding for BFB, while TRP210 and VAL245 contribute significantly lower energy binding. The unavailable residues (due to truncation) in the ER-α36 LBD active pocket (which lie within the ER-α66 protein binding site) include GLY521, HIS524, LEU525, ASN532, VAL533, VAL535, PRO535, and LEU539.

The energy contributions in kcal/mol by the ER-α36 pocket residues to fulvestrant binding are as follows: MET170 (−1.0), LEU173 (−1.62), ASP178 (0.31), LEU211 (−2.39), MET215 (−1.48), LEU218 (−1.01), and LEU255 (−0.81). The interactions revealed within the FULV-bound ER-α36 LBD are lower in number and energy values compared to the BFB-bound ER-α36 LBD system. At the full-length-ER-α66-containing binding site, BFB interacts with MET343, LEU346, THR347, ALA350, MET421, ILE424, LEU428, LYS531, and VAL533. The energy contributions in kcal/mol by the ER-α66 pocket residues are as follows: MET343 (−1.25), LEU346 (−2.0.), THR347 (−2.45), ALA350 (−1.7), MET421 (−0.8), ILE424 (−0.5), LEU428 (−0.27), LYS531 (−0.11), and VAL533 (−2.12). Despite the additional interactions from the LYS351 and VAL533 residues with BFB at the ER-α66 binding site, the contributing energies are lower than those from the novel interactions by the binding residues in the BFB-bound ER-α36 complex.

In the BFB-bound ER-α36 complex, the negatively charged GLU180 residue contributes significantly more electrostatic energy (−20.38 kcal/mol) but the lowest total energy compared to the residues in the three complexes. The GLU180 COO^−^ group interacts with the two hydroxyl oxygen (O) atoms of 2-(3,4-dihydroxyphenyl) substituent adjacent to the 4H-chromen-4-one group of BFB, forming two conventional hydrogen bonds that contribute large electrostatic force and polar solvation energies (Figure 7C). Similarly, the negatively charged ASP178 contributes to the electrostatic and polar solvation energies in the FULV-bound ER-α36 complex by forming an H-bond with an O atom on the FULV phenyl ring. Whereas, in the BFB-bound ER-α66 complex, the THR347 backbone interacts with an O atom on the 2-(3,4-dihydroxyphenyl) group of the BFB to form a conventional H-bond, and the ALA350 backbone interacts with an O atom lying on the 4H-chromen-4-one parent group of the BFB. However, this interaction was not observed with the BF-bound ER-α66 complex. All three of the ligand-bound complexes display hydrophobic interactions. The following residues involve hydrophobic interactions in BFB-bound ER-α36, thus contributing to the overall van der Waals force: MET170 (MET343), LEU173 (LEU346), ALA177 (ALA350), TRP210, LEU211, LEU214, MET215, LEU218, PHE231, VAL245, and MET248 (MET421). The residues in brackets are the ER-α66 residues equivalent to the ER-α36 residues, which form hydrophobic forces with ER-α66. The other residues that form hydrophobic interactions with BFB in the ligand-bound ER-α66 but do not interact in the bound ER-α36 include LEU349, VAL418, MET421, ILE428, LYS531, and VAL533.

For the BFB-bound ER-α36, the sidechain phenyl group of PHE231 forms a pi-hydrophobic (pi-pi T-shaped) interaction with the phenyl substituent adjacent to the parent group of BFB; MET170, LEU211, LEU214, VAL245, and MET248 form alkyl hydrophobic interactions with the 6, 8-bis(3-methylbut-2-en-1-yl) group of the BFB; and LEU173, ALA177, TRP210, and LEU218 form mixed pi/alkyl (pi-sigma and pi-alkyl) hydrophobic interactions. In addition, the sulfur atom of MET215 forms a pi-sulfur interaction with the BFB’s 2-(3,4-dihydroxyphenyl) group. MET170, LEU173, LEU211, MET215, LEU218, and LEU255 form alkyl hydrophobic forces with the long aliphatic chains of fulvestrant in the FULV-bound ER-α36 LBD protein. For the BFB-bound ER-α66 LBD system, Met343, MET421, ILE424, LEU428, and LYS531 form alkyl hydrophobic interactions with the 6, 8-bis(3-methylbut-2-en-1-yl) group of the BFB, while LEU346, THR347, ALA350, and VAL533 form pi-alkyl interactions with the adjacent 3,4-hydroxyphenyl group and parent scaffold 4H-chromen-4-one.

#### 3.3.3. Comparative Stability and Flexibility of Estrogen Receptor Variants

In order to further investigate the changes in the secondary structure of the ER-α36 LBD and ER-α66 LBD, we computed the root mean square deviation (RMSD) of the unbound and bound estrogen receptor variants. The RMSD of the C-α atoms of the amino acid backbones is used as an indicator of the stability of a protein structure. The RMSD can estimate the deviations occurring in the backbone atoms of a protein during molecular dynamics simulations. A high RMSD value implies increased atomistic deviations, consequently revealing an unstable protein structure. At the same time, a low RMSD value correlates with a decreased atomistic deviation, implying a stable structural protein system.

Figure 10 shows the binding of the BFB-induced conformational alterations in the ER-α36 LBD and the ER-α66 LBD across the 200-ns MD simulation time and the reference inhibitor (FULV)-induced structural changes in the ER-α36 LBD. The evidence is seen with the conspicuous deviations within the residues’ backbone atoms of the BFB-bound ER-α LBD variant proteins compared to its unliganded form. In addition, averaged RMSD (Å) values of 2.147 and 2.024 were computed for the apo ER-α36 LBD and apo ER-α66 LBD, respectively, indicating that the ER-α66 LBD is more compact than the ER-α36 LBD. The BFB-bound ER-α36 and FULV-bound ER-α36 LBD showed higher RMSD (Å) values of 2.089 and 2.273, respectively. At the same time, 2.039 Å was estimated as the RMSD value of the BFB-bound ER-α66 LBD complex. The BFB-bound ER-α66 LBD is a more compact bound enzyme among the bound systems. Although the difference in the RMSD values of the bound and unbound systems is less than 1.0 Å, Figure 11A–D (visual snapshots) indicates significant structural alterations upon the binding of the inhibitors to either of the estrogen receptor variants. The estrogen receptor LBD structures become more stable upon binding to the BFB, unlike when fulvestrant binds to the ER-α36 LBD. However, the ER-α36 LBD was shown to be less stable compared to the ER-α66 LBD.

Moreover, the higher stability of both the unliganded and the bound ER-α66 LBD than the ER-α36 LBD could be due to LYS531 and VAL533, which are absent in the ER-α36 partial LBD. These alterations agree with the previous studies and can impact estrogen receptor signaling and expression in breast cancer initiation and development [17,18]. The RMSD trajectories also showed that the ER-α36 partial LBD pocket converged at 20 ns, compared to the ER-α66 LBD, which converged at about 30 ns, and the difference in their fluctuations is no greater than 0.65 Å (Figure 11).

Figure 11A–D illustrates the binding poses of BFB and FULV at the active site of the ER-α36 LBD and the ER-α66 LBD compared to the unbound systems at 200 ns. Furthermore, we investigate the fluctuation behavior of the estrogen receptors by estimating the RMSF value, which indicates the mobility of every residue in the protein structure. In comparison to fulvestrant at the ER-α36 LBD site, the RMSF plot (Figure 10B) of broussoflavonol B at both the ER-α36 truncated LBD and the ER-α66 complete LBD site shows a more significant displacement of the residues. The mean RMSF values of the unbound ER-α36 and ER-α66 systems were 1.260 Å and 0.848 Å, respectively. Moreover, the higher RMSF value of the BFB-bound ER-α36 was critically impacted by the residue GLU180 (Figure 10B). More significant movement of all of the residues was seen when BFB bound to the novel truncated ER-α36 LBD (mean RMSF = 1.182 Å) compared to the BFB at the site of the complete ER-α66 LBD (mean RMSF = 1.037 Å). Unlike the ER-α36 site that showed decreased RMSF upon BFB and FULV binding, ER-α66 showed increased fluctuation when BFB bound to its pocket. There is not much flexibility difference in either of the ER-α binding sites, except in the ER-α66 (residues: 330–340 and 528–540) and in the FULV-bound ER-α36 (residues: 190–250), where slightly larger fluctuations occurred.

#### 3.3.4. Analysis of RoG, PCA, and DSSP

To gain further insights into the comparative characteristic 3D conformations of the studied ER-α variants, we computed the radius of gyration (RoG), principal component analysis (PCA), and the defined secondary structure of the protein (DSSP) for the liganded and unliganded systems. The C-α radius of gyration (RoG) measures the structural compactness at the binding interfaces of ER-α36 and ER-α66, in which a less tightly packed protein structure resulted, due to a high RoG value and increased mobility. In contrast, a low ROG value indicates more tightly packed estrogen receptor conformation and reduced mobility.

Figure 12A shows the RoG plots of the ligand-bound ER-α36 systems. The unbound estrogen receptor variants showed diminished RoG values compared to their corresponding bound complexes.

The average RoG (Å) of the BFB-bound ER-α36, FULV-bound ER-α36, apo ER-α36, BFB-bound ER-α66, and apo ER-α66 systems were 17.338, 17.239, 17.399, 18.686, and 18.588, respectively. The values imply no significant difference between the bound and unbound respective estrogen receptor variants. However, the bound ER-α36 showed decreased RoG values, while the bound ER-α66 showed increased RoG values when each variant was compared to their respective apo structures. Clearly, the BFB-bound ER-α66 showed a significant RoG difference compared to the BFB-bound ER-α66, which indicates that the ER-α36 protein is a more tightly packed system with reduced mobility. In general, the ER-α36 protein showed a diminished and lower RoG value than ER-α66, indicative of a more tightly packed and less mobile protein. These findings further justify the above-estimated flexibility results from the RMSF analysis, in which a more tightly packed and reduced mobility structure should fluctuate less than a less tightly packed and increased mobility system.

Figure 11A–D elucidates the visual motion shifts across two principal components for the unbound and ligand-bound systems. PCA is a critical analysis for studying simulation trajectories over a time range. The first two principal components (PC1 and PC2) were computed from the 200-ns MD trajectories of the ER-α protein C-α atoms, and the conformational patterns of the bound and unbound receptor variants were projected along the first two eigenvectors (ev1/PC1 versus ev2/PC2) using the C-α atoms’ Cartesian coordinates. The eigenvalues represent variance magnitude, and the eigenvectors indicate variance direction. The PCA scatter plots were created using Origin 6.0. The eigenvectors computed from the MD trajectory for all of the systems vary, indicating the protein motion difference between the two ER-α variants. The most significant fluctuation modes with the bound and unbound ER-α protein systems’ motion were determined, as shown in Figure 12B.

The conformational behaviors of the bound and unbound ER-α protein variants were projected directionally along PC1 and PC2 to gain insights into the separation of their motions. The PCA plots showed a distinct motion separation in the essential subspace along both principal components among the five ER-α protein systems, with the bound complexes exhibiting highly dispersed motion compared to the unbound proteins. This significant difference is evident from the characteristic structures along the two components. The unbound ER-α36 variant appears to be more dispersed than ER-α66 (more compact motion), concentrated at the center of the two eigenvectors. In straightforward terms, the binding of BFB to either of the ER-α variants showed highly dispersed motions. The induced dispersions in the BFB- and FULV-bound systems could indicate active conformations in the protein, which were not seen in the apo ER-α66 protein, due to compactness and immobility. Inferentially, the active conformation of the protein, such as the whole ER-α and the pocket landscape, could weaken the binding affinity of the ligand.

Therefore, these findings further reflect that the binding of BFB and FULV stabilizes a highly active protein conformation favoring ligand binding, which corroborates the above-explained analytical results. The high ER-α inter-residual displacement induced by the BFB tumor might have resulted in the cell cycle arrest reported in the experimental wet lab studies.

#### 3.3.5. Analysis of DSSP, Distance, and Torsion Angles of the ER-α36 Variant

Furthermore, a comparative defined secondary structure protein (DSSP) analysis was performed on the 200-ns trajectories of the bound and unbound whole ER-α36 and ER-α66 protein systems in order to monitor the conformational changes in their secondary structures. The prior MD and the 200-ns MD structures of the unbound and bound ER-α36 revealed significant structural differences. Figure 13 shows some visual changes in the ER-α36 protein systems, comparing the prior-MD, post-MD, unbound, BFB-bound, and FULV-bound systems. Changes such as twisting, coiling, shifting, and secondary structure transition from one element type to another were observed, particularly in the BFB-bound ER-α36 (green).

The visual observations of the loop (proximal to the active site) LYS243, CYS244, VAL245, GLU246, GLY247, MET248, VAL249, GLU250, and ILE251 showed that it shifted and twisted to cover the unbound ER-α36 system at 200 ns (deep pink). Likewise, the BFB-bound ER-α36 loop that contains residues SER283, GLY284, PHE285, THR286, ILE287, SER288, HIE289, VAL290, and GLU291 and the loop that contains residues LYS189, ARG190, VAL191, PRO192, GLY193, PHE194, VAL195, ASP196, LEU197, THR198, LEU199, and HIE200 were visually observed to pull toward the binding pocket in the presence of broussoflavonol B.

Further visual inspections showed that LEU201, LEU202, GLU203, CYS205, ALA206, ILE213, LEU214, MET215, MET248, VAL249, and GLU250 of the unbound ER-α36 transitioned into the loop, while the loop structure of LEU133, ALA134, and LEU135 and SER136 of the apo ER-α36 at 200 ns transitioned into the helices. Due to the interactions of BFB with two helical structures at the active site, these changes caused significant conformational changes in the ER-α36 compared to the remaining systems. In contrast, LEU199, HIS200, VAL290, and GLU291 of the BFB-bound ER-α36 transformed from the helix into the loop.

The binding of a ligand to a protein often induces structural changes to said protein. These conformational changes were further investigated by computing comparative defined secondary structures of the systems (Figure 14A–E). The structural assignment of the systems involves the prominent secondary structure elements, including alpha helices, beta sheets, loops, and coils, which can be sub-categorized as bend, alpha, helices (pi and 3_10_), turn, anti, para, and none structures. The figures show the positions of each element present in the apo and bound ER-α36 and ER-α66 structures during the 200-ns MD simulation.

Figure 14A,B show the DSSP plots for unbound ER-α36 and ER-α66 systems, while Figure 14C–E represent the bound complexes BFB–ER-α36, BFB–ER-α66, and FULV–ER-α36, respectively. As shown in the plots, there are similarities in the secondary structures of ER-α36 and ER-α66, especially the alpha, para, and bend structures. However, we can observe significant differences, particularly changes in the residues 1–16, 31–36, 76–81, and 155–181. The number of residues transitioning from one secondary structure to another is vital between the unbound proteins and compared with the bound systems. These findings may account for the stability and flexibility discussed in Section 3.3.3, especially comparing the notable changes between the unbound and bound ER-α36 and the conspicuous structural differences between the unbound and bound ER-α66. Therefore, these insights could aid in designing potential small-molecule inhibitors against cancer tumors.

Furthermore, we have comparatively investigated the conformational changes and space of the whole and binding pockets of the truncated ER-α36 LBD and completed ER-α66 LBD snapshots seen in Figure 15A–E. The distance and torsion angle geometric approaches were used to investigate four of the residues of the estrogen receptors that are critically significant to the binding affinity of the BFB and FULV. However, the bond length and torsion angles of the four residues can be extrapolated to the geometry of the remaining residues, because reports have shown that slight variations occur among the residues of biomolecules.

Protein spontaneously samples the conformational space by variations in torsional angles, significantly impacting the lower energy barrier. Torsion angles are the natural degrees of freedom of protein structures, which provide an understanding of the mechanisms of functional conformational changes or computational models of protein dynamics. Moreover, torsion angles describe the biologically relevant quantities better than atomic coordinates (*x*, *y*, *z*), and the geometric relation of two parts of a molecule can be joined, though it is the most uncomplicated method used to define atoms’ position that does not naturally sample the conformational space.

Accordingly, the predicted torsion angles can accelerate protein folding and structure prediction to improve structural precision. However, the ease of obtaining torsional variations that correspond to observed changes in Cartesian coordinates is helpful in protein studies. In addition, the torsion angle of a residue’s backbone can impact all of the other protein angles and influence the protein conformation upon ligand binding.

Therefore, the torsion angles of the C-α of LEU173, GLU180, LEU214, and PHE231 were computed in this study. Figure 14 shows the ligands binding to the active sites of the ER-α LBD proteins compared to the unliganded protein structures. The ER-α36 LBD C-α atoms’ interacting residues involve hydrogen and oxygen bonding and include LEU173, GLU180, LEU214, and PHE231, with the following corresponding ER-α66 LBD residues: LEU346, GLU353, LEU387, and PHE404, respectively. The 4H-chromen-4-one and 2-(3,4-hydroxyphenyl) of the broussoflavonol remain buried within the hydrophobic active site of the ER-α36 and ER-α66 proteins, while the aliphatic end remains on the surface of the pocket. In contrast, the fulvestrant fluorine atoms protrude into and interact with the hydrophobic pocket of the ER-α36 LBD.

The visual observations of the MD trajectory snapshots show that the conformation of the unbound ER-α LBD active sites is characterized by twisting and the transitional recoiling of the loop covering the binding cavity. Therefore, we computed the distance and angle metrics to accurately describe the motions in order to understand the binding landscape of the estrogen receptors instead of using the distance metrics alone. We have described the dynamics and motion of the comparative binding pockets of the estrogen receptors using the distance and TriC-α angles’ (*θ*, *ϕ*) combination. Here, Figure 16 illustrates the parameters used to describe the motions, considering the *distance* (*d*) between the two critical pocket residues (GLU180 and PHE231) relative to *angles θ* (LEU173, GLU180, and LEU214) and *ϕ* (LEU173, GLU180, LEU214, and PHE231). These computational parameters have been extensively reported as a better option to describe the motion of the protein flaps and dimers in many dimensions, even in one of our previous works.

Table 4 confirms the estimated distances between the critical LBD residues GLU180 and PHE231 of the ER-α36 LBD and the estimated distances between their corresponding residues in the ER-α66 LBD. The distances, *d*_0_ (prior to the MD simulations), between the active site residues GLU180 and PHE231 of the unbound enzymes ER-α36 and ER-α66 were 7.743 Å and 7.406 Å, respectively. Both systems had partial openings at 1 ns, but ER-α36 showed a full further pocket opening at 10 ns (*d*_10_ = 5.882 Å) as it opened very wide (*d*_10_ = 11.170 Å). However, the ER-α36 active pocket opened again at 50 ns, and the ER-α66 pocket opened at the same simulation time. After another 50-ns MD simulation, we observed that the two pockets closed by shifting to the same positions observed at the first 50 ns. Therefore, both the ER-α36 and the ER-α66 binding activities could be said to undergo opening and closing conformations. However, ER-α36 showed a tighter binding pocket than ER-α66, as evident from the distances at the simulation time events (Table 5).

Interestingly, this is the first work that has observed the asymmetric opening of the ER-α36 and ER-α66 active pockets. The asymmetrical opening of the pockets is characterized by various structural transitions and even coiling and twisting, corresponding to a significant shift in the torsional angle of ER-α36 and the ER-α36 at various simulation time ranges. These insights describe the intensive movements of the interacting residues buried within and proximal to the binding cavities, which might create the steric hindrance that causes the opening of the active site.

Furthermore, Table 5 shows the angle *θ* and dihedral angle *ϕ* of the bound and unbound ER-α36 and ER-α66 pocket residues, whose impact can be extrapolated to the whole enzyme at a time range of 1–200 ns. It is evident that the distance is inversely proportional to the angle *θ* metrics but increases as *ϕ* increases or decreases as *ϕ* decreases. Overall, there seems to be a large shift in the dihedral angle (*ϕ*) of the bound and unbound ER-α66. In the bound estrogen receptors, LEU173, GLU180, LEU214, and PHEE231 tightly embraced the ligand BFB without a significant fluctuation from the residues.

Without a doubt, this study will ultimately contribute to designing potent structure-based inhibitors against estrogen receptor variants, especially ER-α36, by targeting the four residues and critical secondary structures, such as the loop covering the cavity. The design will be carried out to inhibit the opening and closing of the active sites.

The computational metrics used in this study provide insights into the inducing effect of BFB by unravelling the molecular mechanism of action of the compound to corroborate the experimental cytotoxic effects (IC_50_) of BFB at the Gap_0_ and Gap_1_ phases reported by Jeong and Ryu [18] and Guo et al. [17]. Put together, the significantly higher binding-free energy of BFB upon the binding to ER-α36 (compared to FULV-ER-α36 and BFB-bound ER-α66) and various ER-α36 conformational changes, as given by RMSD, RMSF, RoG, PCA, and DSSP, indicate the impact of BFB on the cell cycle activity. Therefore, these findings provide a model that describes the mechanisms by which the BFB compound induces downregulation-accompanied cell cycle arrest at the Gap_0_ and Gap_1_ phases when it binds to the ER-α36 protein.

## 4. Conclusions

The inhibition of negative breast cancers by downregulating the estrogen receptor-α36 ligand-binding domain (LBD) with a few inhibitors (e.g., broussoflavonol B) and FDA-approved drugs (e.g., tamoxifen and fulvestrant) has been reported extensively. Nevertheless, there is no crystallized structure of ER-α36, and the dynamics and motions of the secondary or tertiary structure have been reported. The defining of the binding landscape of the ER-α36 LBD using the computational parameters provides a platform for understanding the drug resistance caused by the mutations. Furthermore, unravelling the mechanism of inhibition of the ER-α36 active site justifies the importance of this study. This study will significantly assist in the design of potent inhibitors to bind to the flexible LBD cavity.

Conclusively, the homology model of the ER-α36 enzyme allows us to gain insights into the active structures of the protein. The predicted protein’s LBD structure shows unique features, such as the number of helices, strands, loops, and the 27 C-terminal amino acids. It also unfolds some significant similarities with its variant ER-α66. Moreover, the active site of both enzymes has about 80% conserved identity, and more than 95% of the predicted binding cavity residues occurred in the favored region. The docking binding affinity (BA) and the MMGB(PB)SA free binding energy (ΔG) of the BFB at the truncated pocket of the ER-α36 are significantly higher than those of the ER-α66 and FULV docked at the ER-α36 pocket.

Moreover, the ΔG of the BFB at the ER-α66′s pocket is significantly higher than the ΔG of the FULV at the pocket of the ER-α36. Therefore, the estrogen receptor variant (ER-α or ER-α66) can still be an explorable target for a cancer drug strategy. Although drug resistance against fulvestrant has been reported, the FDA-approved drug has been shown to have unavoidable high BA and ΔG values, indicating the probable downregulation or inhibition of the signaling activity of ER-α36.

The adapted combined computational metrics, including distance, TriCα, and dihedral angles, unfold critical motions relating to the structure, binding pocket, and mechanism of inhibition of the enzymes’ functions. The dynamics and motions of the ER-α36 binding cavity have revealed flexible asymmetric opening and closing conformation. The results presented in this study on the dynamics and motion behaviors of the active conformation of the estrogen enzymes (ER-α36 and ER-α66) and the description of their binding cavity landscape will undoubtedly enhance the designing of potent structure-based compounds that will scale through the clinical trial of cancer drug development.

## Figures and Tables

**Figure 1 biomolecules-13-01798-f001:**
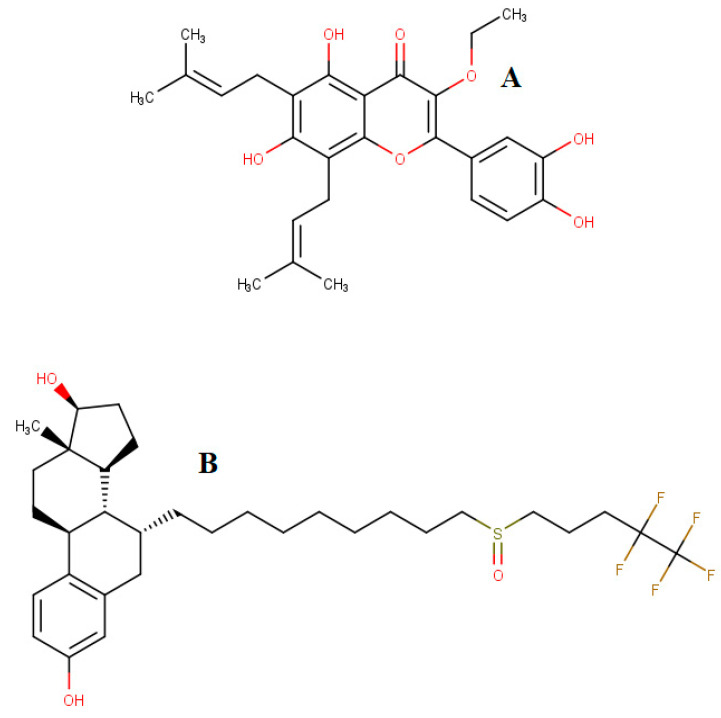
Molecular structures of broussoflavonol B (**A**) and fulvestrant (**B**).

**Figure 2 biomolecules-13-01798-f002:**
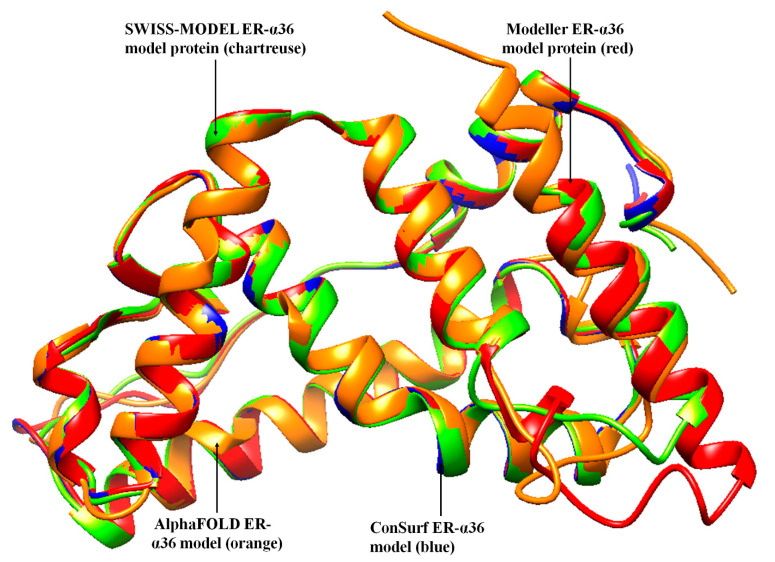
Comparative superimposition of the four ER-α36 models by the SWISS-MODEL, Modeler, AlphaFOLD, and ConSurf databases.

**Figure 3 biomolecules-13-01798-f003:**
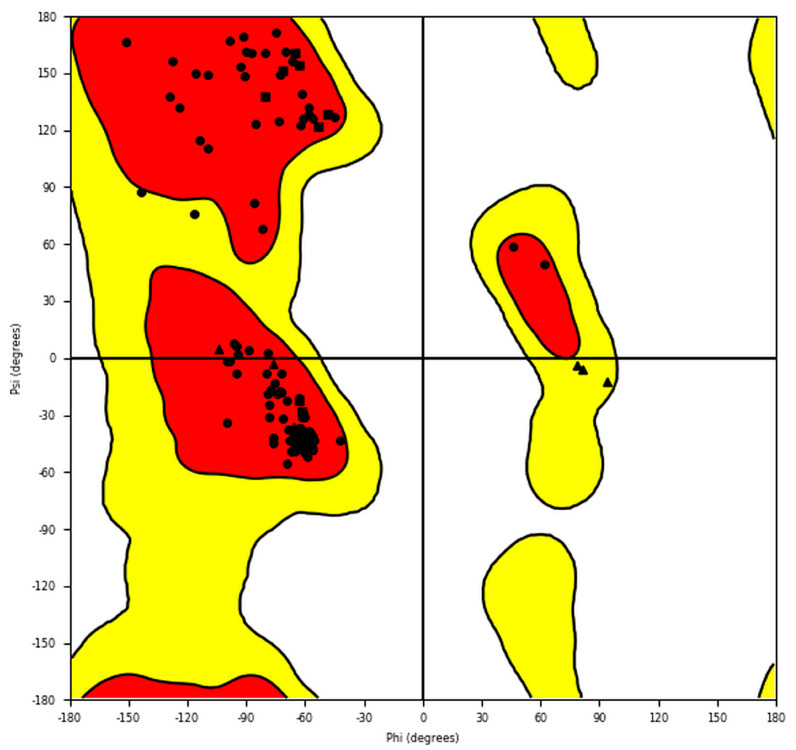
The Ramachandran plot for the homology model of the amino acid sequence of human ER-α36 protein. The red and yellow colours represent the most favoured and allowed regions. The red regions correspond to conformations where there are no steric clashes in the model ER-α36, including the dihedral angles typical of alpha-helical and beta-sheet conformations.

**Figure 4 biomolecules-13-01798-f004:**
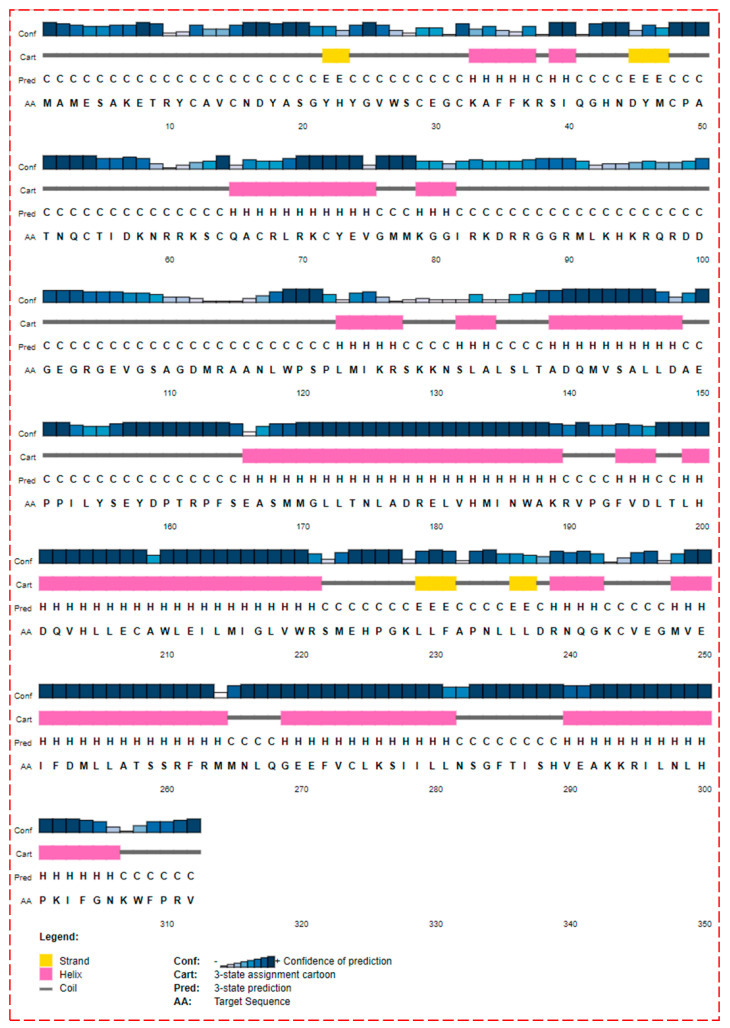
Psipred prediction of the secondary structure of the estrogen receptor-α36.

**Figure 5 biomolecules-13-01798-f005:**
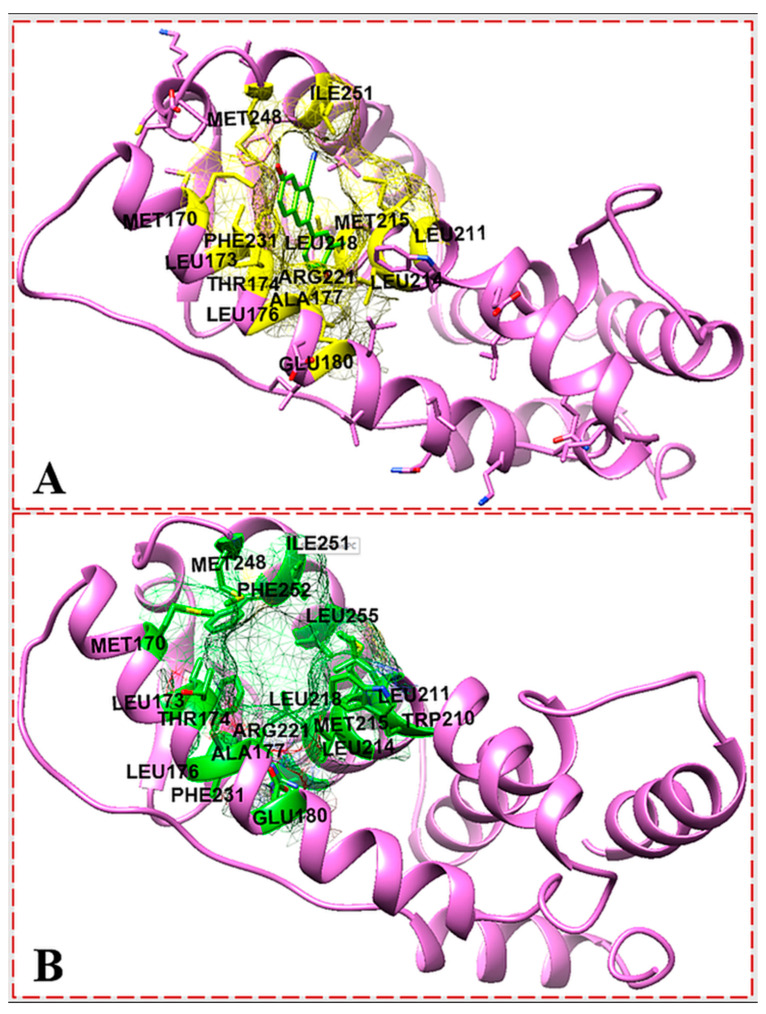
IntFOLD server predicted active site (**A**); Castp server identified active site (**B**); showing the active site residues of ER-α36 protein.

**Figure 6 biomolecules-13-01798-f006:**
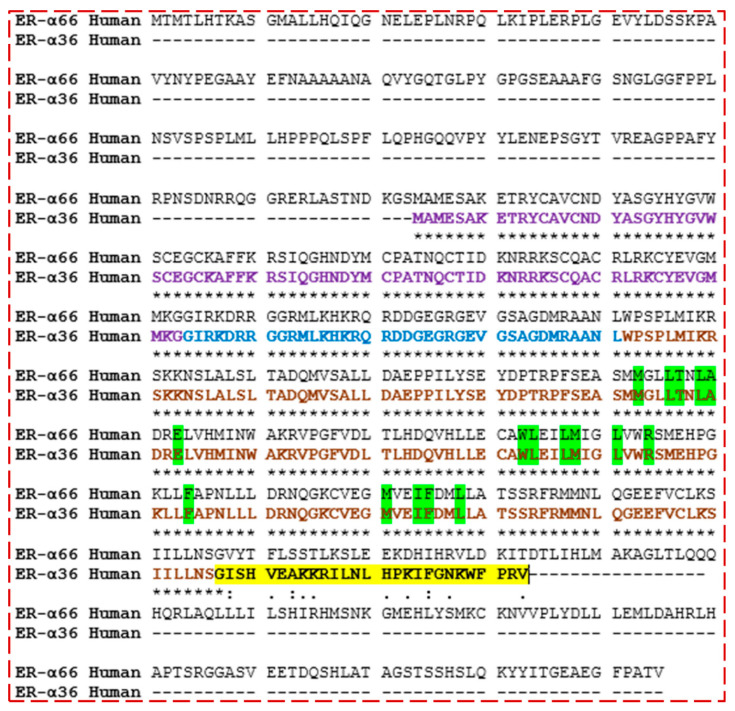
Pairwise sequence alignment of ER-α36 and ER-α66 proteins showing the degree of amino acid conservation, including the unique active sites (green), DNA-binding domain (purple), Hinge domain (blue), ligand-binding domain (LBD) (brown), and 27 unique amino acids (yellow, NOT found in ER-α66 LBD). Fully identical amino acid observed (*), Observed conserved substitutions in the amino acid sequences (**:**), and Semi-conserved substitutions observed (**.**).

**Figure 7 biomolecules-13-01798-f007:**
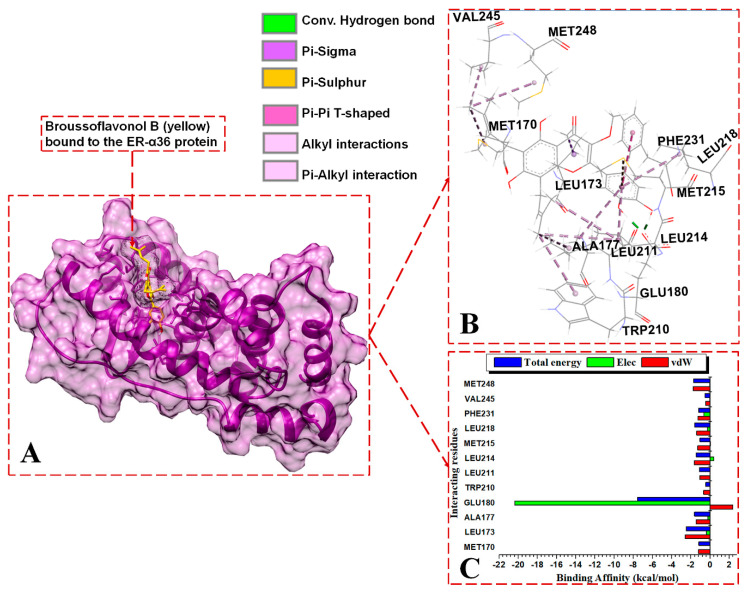
Per-residue energy contributions to the binding of BFB to ER-α36 LBD. (**A**) BFB binds to the active site of the ER-α36, (**B**) interaction network of the BFB-ER-α36 complex, and (**C**) per-residue energy decomposition to BFB.

**Figure 8 biomolecules-13-01798-f008:**
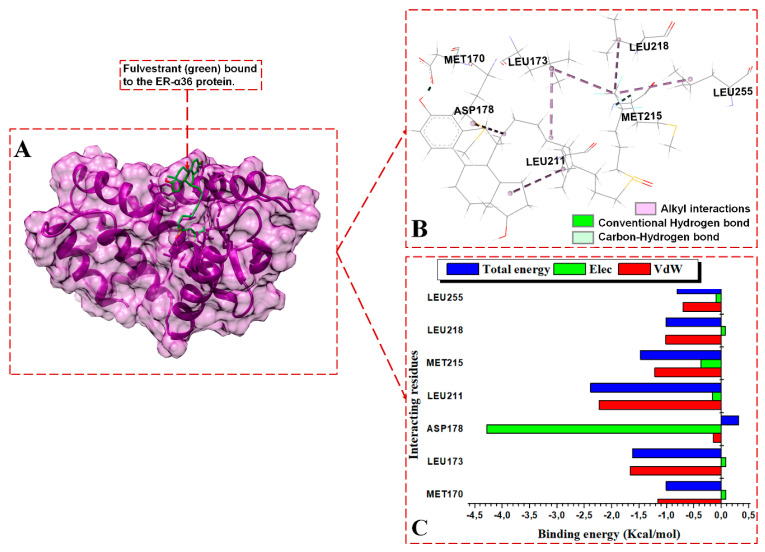
Per-residue energy contributions to the binding of FULV to ER-α36 LBD. (**A**) FULV binds to the active site of the ER-α36, (**B**) interaction network of the FULV-ER-α36 complex, and (**C**) per-residue energy decomposition to FULV.

**Figure 9 biomolecules-13-01798-f009:**
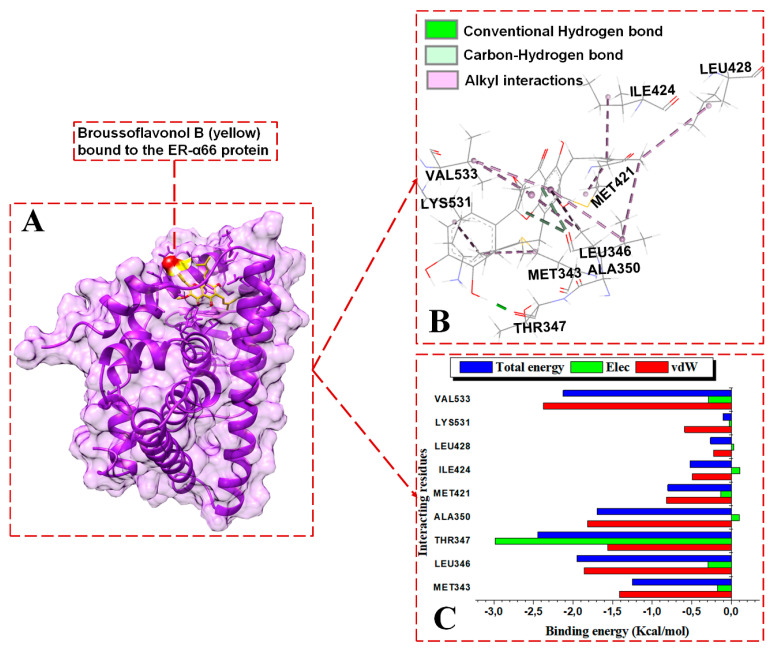
Per-residue energy contributions to the binding of BFB to ER-α66 LBD. (**A**) BFB binds to the active site of the ER-α66, (**B**) interaction network of the BFB-ER-α66 complex, and (**C**) per-residue energy decomposition to BFB.

**Figure 10 biomolecules-13-01798-f010:**
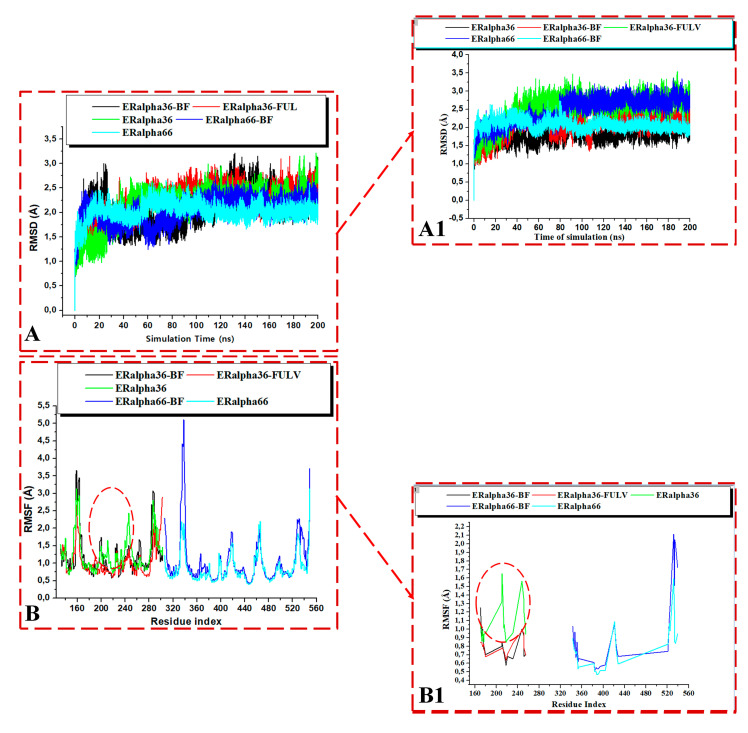
RMSD plots of the whole (**A**) and the active site (**A1**) for the Apo and whole ligand-bound ER-α36 LBD and ER-α66. RMSF plots of the whole (**B**) and the active site (**B1**) for the Apo and ligand-bound ER-α36 LBD and ER-α66. BFB-bound ER-α36 (black), FULV-bound ER-α36 (red), ApoER-α36 LBD (green), BFB-bound ER-α66 (blue), and ApoER-α66 LBD (cyan).

**Figure 11 biomolecules-13-01798-f011:**
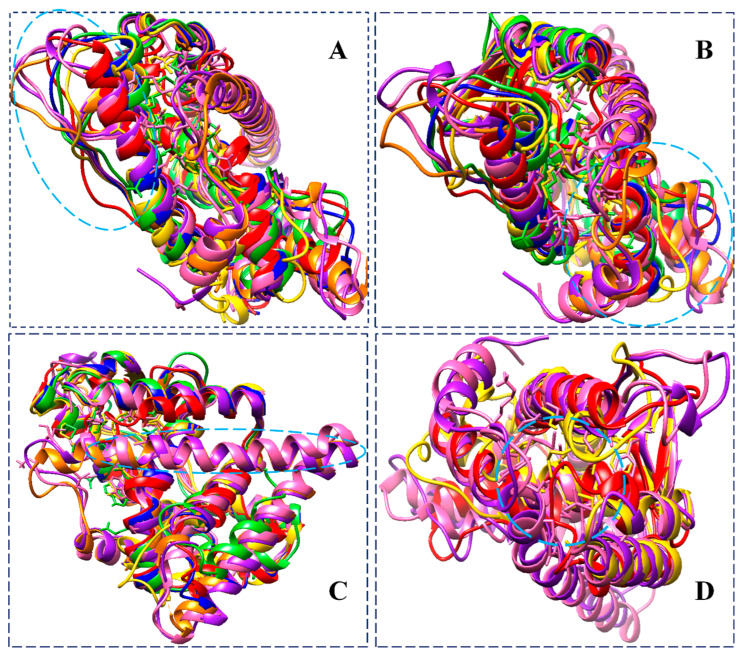
Superimposition of unbound ER-α36 LBD (blue) pre-MD, unbound ER-α36 LBD (red) at 200 ns, BFB-bound ER-α36 LBD (gold) at 200 ns, FULV-bound ER-α36 LBD (lime green) at 200 ns, unbound ER-α66 LBD (orange) pre-MD, ER-α66 (purple) at 200 ns, and BFB-bound ER-α66 LBD (hot pink), showing distinct conformational changes. Insets: active site proximal loop extended to α_2_-helix (**A**,**B**). ER-α66 LBD Helix-12 not available in the ER-α36 LBD. The helix-12 of the ER-α66 and (**C**) moved to cover the active site of the ER-α66 (**D**).

**Figure 12 biomolecules-13-01798-f012:**
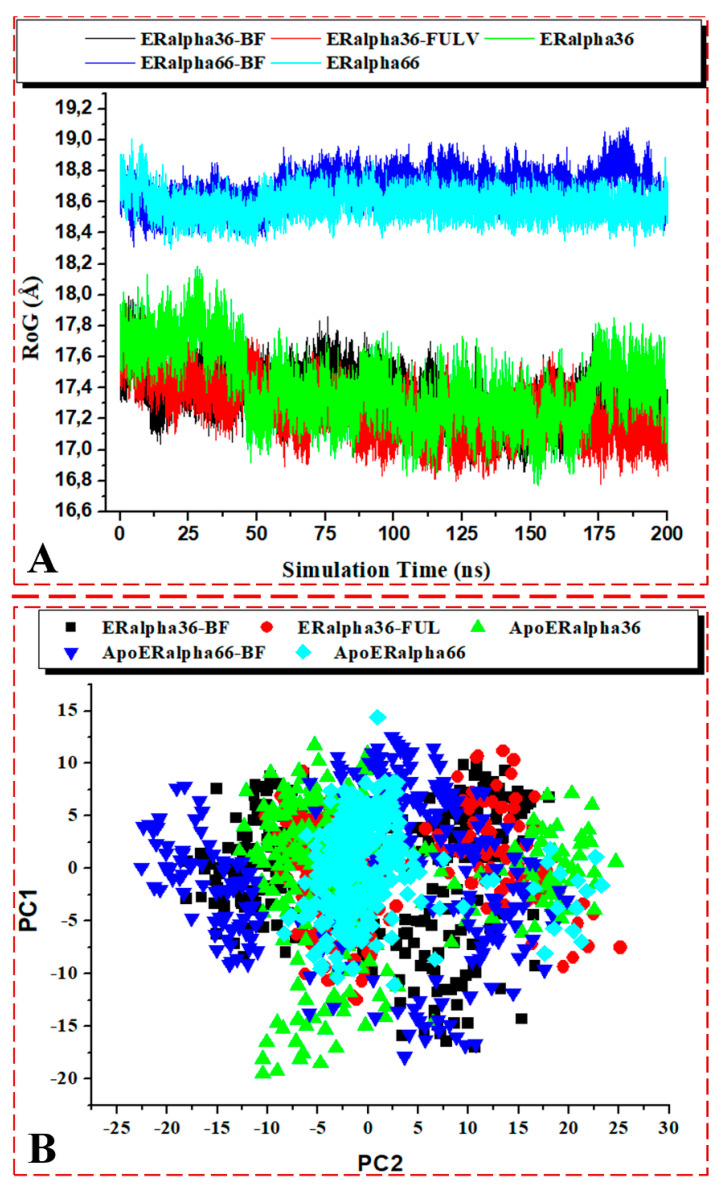
(**A**) C-α atoms’ RoG and (**B**) PCA scatter plots of 2000 snapshots along the first two principal components of the ER-α36 LBD and ER-α66 LBD systems. ER-α36–BFB (black), ER-α36–FULV (red), unbound ER-α36 (green), ER-α66–BFB (blue), and bound ER-α66 (cyan). PC1 and PC2, respectively, show differences in motion among the bound and unbound ER-α variants over the simulation time.

**Figure 13 biomolecules-13-01798-f013:**
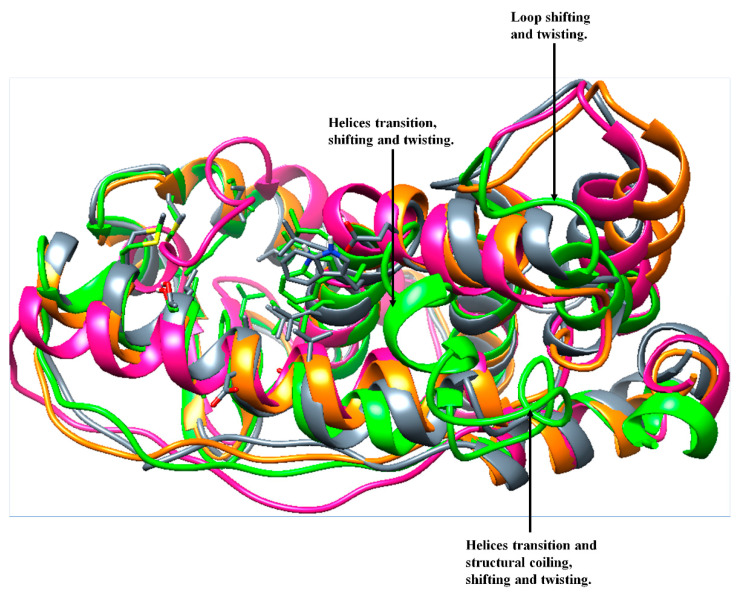
Visual secondary structure changes: superimposed structures of the prior-MD Apo ER-α36 (orange), 200 ns of Apo ER-α36 (deep pink), BFB-bound Apo ER-α36 (green), FULV-bound ApoER-α36 (slate gray), Apo ER-α66 (dark cyan), and BFB-bound ER-α66 (purple) systems.

**Figure 14 biomolecules-13-01798-f014:**
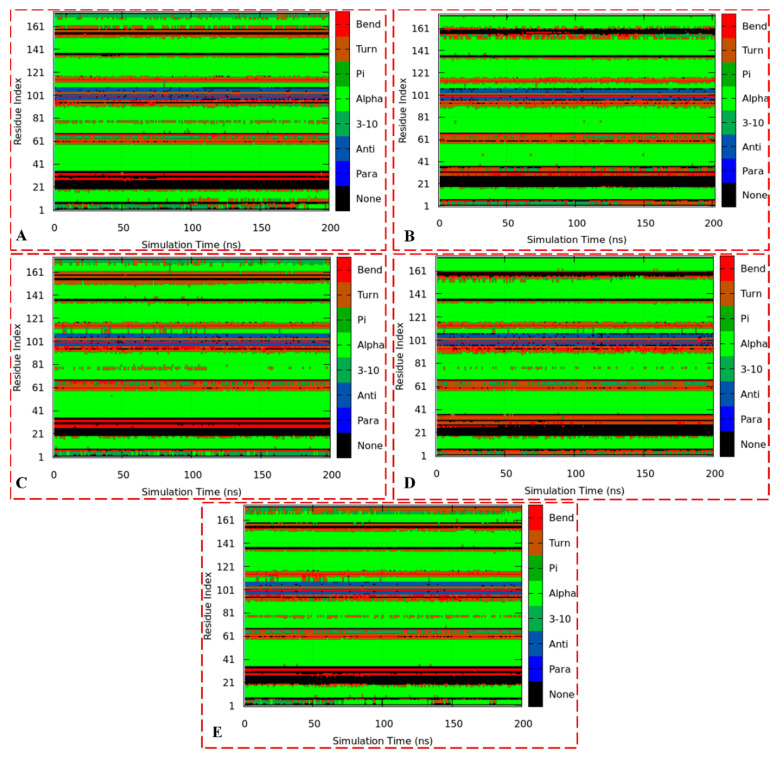
The trajectory DSSP plots of the apo ER-α36 LBD (**A**), apo ER-α66 LBD (**B**), BFB-bound ER-α36 LBD (**C**), BFB-bound ER-α66 LBD (**D**), and FULV-bound ER-α36 LBD (**E**).

**Figure 15 biomolecules-13-01798-f015:**
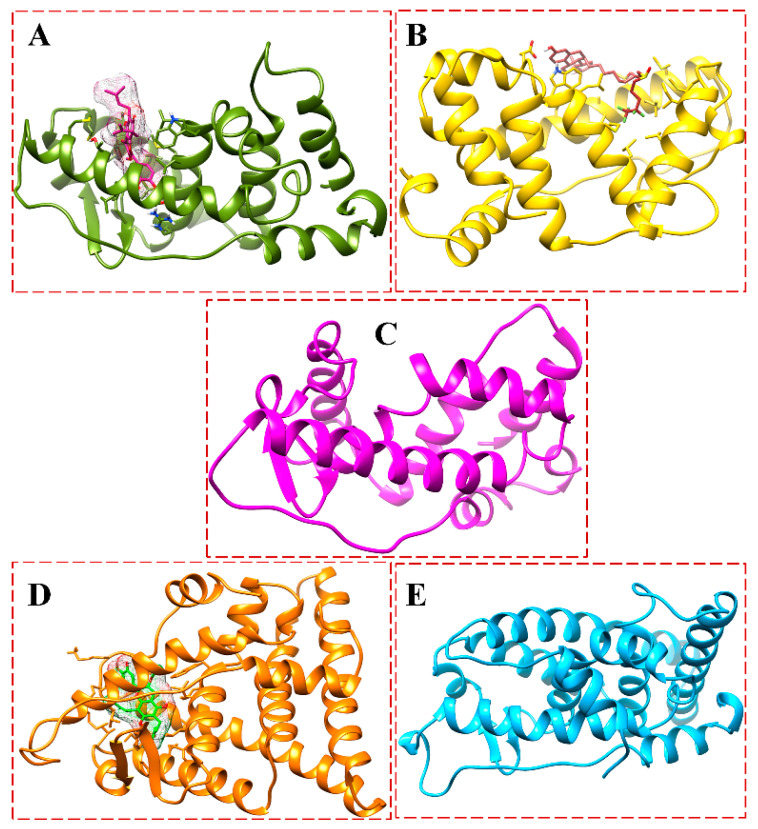
Snapshots at 200-ns MD simulations of apo and complexes: ((**A**), dark green) BFB-bound ER-α36 LBD, ((**B**), gold) FULV-bound ER-α36 LBD, ((**C**), deep pink) unliganded ER-α36 LBD, ((**D**), orange) BFB-bound ER-α66 LBD, and ((**E**), deep sky blue) unliganded ER-α66 LBD.

**Figure 16 biomolecules-13-01798-f016:**
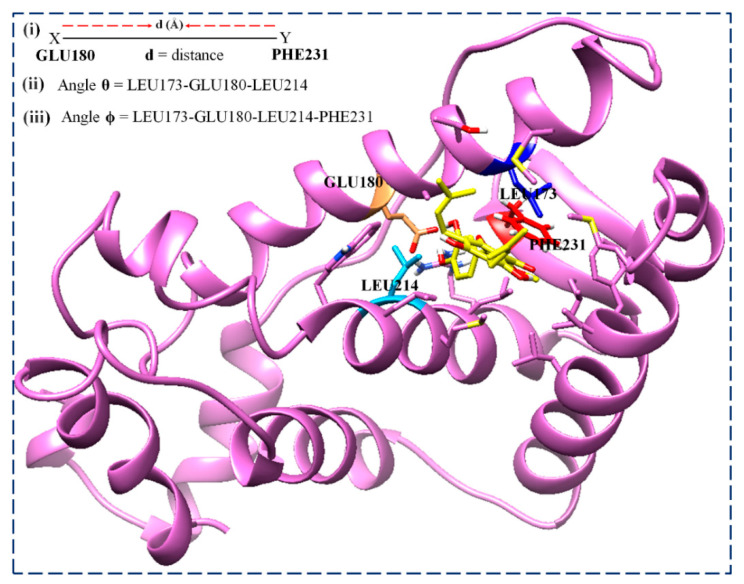
Comparatively defining the binding landscape of the ER-α LBD variants using the distance and angle metrics. Residue color: LEU173 (blue), GLU180 (sandy brown), LEU214 (deep sky blue), and PHE231 (red).

**Table 1 biomolecules-13-01798-t001:** Criteria for choosing the templates used to model ER-α36 in each method.

Species	Database ID	Method	Sequence Identity	Sequence Similarity	GMQE	QSQE	Query Cover
			(%)	(%)			(%)
*Rattus norvegicus*	P06211.1	AlphaFOLD	91.90	0.59	0.80	-	97.0
*H. sapiens*	PDB7prw	X-ray (2.5 Å)	31.18	0.37	0.59	0.24	89.0
				E-value	Max Score	Total Score	
*H. sapiens*	1R5K	X-ray (2.7 Å)	89.78	5 × 10^−115^	333	333	57.0
*H. sapiens*	2P15	X-ray (1.94 Å)	89.73	1 × 10^−114^	332	332	57.0
*H. sapiens*	6DF6	X-ray (2.50 Å)	83.66	6 × 10^−115^	334	334	62.0
*H. sapiens*	POS	ConSurf					

**Table 2 biomolecules-13-01798-t002:** The binding affinity of BFB and FULV on their respective enzymes.

Ligand	Target	Docking Scores (kcal/mol)
Brussoflavonol B	ER-α66	−6.5
Brussoflavonol	ER-α36	−7.7
Fulvestrant	ER-α36	−7.3

**Table 3 biomolecules-13-01798-t003:** Thermodynamics analysis: summary of the MM/GBSA-based binding-free energy involving ER-α36—BFB, ER-α66—BFB, and ER-α36—FULV complexes.

Energy Components (kcal mol^−1^)							
Complex	ΔE_vdW_	ΔE_elec_	ΔG_gas_	E_GB_	E_SA_	ΔG_solv_	ΔG_bind_
ER-α36—BFB	−41.91	38.23	−80.15	34.04	−6.46	27.57	−52.57
	(±4.31)	(±2.02)	(±5.39)	(±3.01)	(±0.43)	(±2.92)	(±4.09)
ER-α36—FULV	−42.64	−12.98	−55.62	24.70	−6.51	18.20	37.43
	(±5.58)	(±7.48)	(±8.59)	(±5.82)	(±0.79)	(±5.64)	(±5.08)
ER-α66—BFB	−48.11	12.65	−60.98	25.28	6.70	18.57	−42.41
	(±5.14)	(±5.25)	(±0.57)	(±5.24)	(±0.57)	(±5.26)	(±4.02)

E_elec_ (electrostatic), ΔE_vdw_ (van der Waals), ΔG_bind_ (calculated total binding free energy), ΔG_gas_ (gas-phase energy), and ΔG_solv_ (solvation-free energy).

**Table 4 biomolecules-13-01798-t004:** Distance, d, between C-α atoms of GLU180 and PHE231 of the ER-α variant systems.

Simulation	Distance (Angstrom)			
Time (ns)	ER-α36	ER-α36–BFB	ER-α66	ER-α66–BFB
0	7.743	7.743	8.817	7.406
1	7.158	7.728	8.784	7.959
10	5.882	8.352	11.170	10.316
50	8.385	8.187	12.493	12.375
100	5.882	8.352	11.170	10.316
150	5.722	8.520	9.782	11.532
200	8.963	7.701	12.041	11.866

**Table 5 biomolecules-13-01798-t005:** Combined TriC-α and dihedral angles relating to simulation time events of the binding cavities of the bound and unbound ER-α66 and ER-α66 enzymes.

Simulation	Bond Angle (θ°)				Torsion Angle (φ°)			
Time (ns)	ER-α36	ER-α36–BFB	ER-α66	ER-α66–BFB	ER-α36	ER-α36–BFB	ER-α66	ER-α66–BFB
0	91.07	65.54	58.65	69.67	−17.33	−22.56	−13.83	−15.57
1	112.46	87.56	82.39	39.63	−25.96	−14.55	−22.86	−32.88
10	53.87	56.86	70.41	89.27	−16.88	−26.99	−30.06	−27.32
50	65.66	54.72	25.98	100.18	−22.55	−30.81	−40.11	−28.56
100	53.87	56.86	70.41	89.27	−16.88	−26.99	−30.06	−27.32
150	75.80	62.57	131.03	103.78	−07.32	−18.45	−40.58	−25.01
200	59.24	60.28	70.93	55.98	−28.21	−16.90	−28.39	−24.22

θ = theta, φ = torsion.

## Data Availability

Data are contained within the article.

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
