# Peer review of "Characteristic Binding Landscape of Estrogen Receptor-α36 Protein Enhances Promising Cancer Drug Design"

_biomolecules, 2023, doi:10.3390/biom13121798_

Round 1
Reviewer 1 Report
Comments and Suggestions for Authors
The authors have undertaken an investigation into the interaction of certain ligands with Estrogen Receptor-α36. It is imperative to underscore that the manuscript exclusively presents in silico data, devoid of any experimental results. It is my contention that, for the successful submission to a high-impact journal, the inclusion of experimental evidence is paramount to substantiate the novel insights under development.
(1) It is noted that the structures depicted in Figure 1 are inverted in accordance with their respective legends. The rationale behind representing the chemical structures with all hydrogen atoms in a 2D model remains unclear.
(2) The legend accompanying Figure 2 is incomplete, merely stating, "Figure 2. Superimposition of…." A comprehensive and informative legend is essential for a complete understanding of the figure.
(3) The Principal Component Analysis (PCA) depicted in Figure 12 requires more thorough discussion and elucidation. The absence of PC1 and PC2 values on the x and y-axis of Figure 2 warrants attention. Additionally, the matrix variables utilized in the analysis are inadequately described.
(4) It is strongly recommended that experimental evidence be incorporated into the results section to enhance the overall quality of the manuscript. This inclusion will fortify the credibility of the findings and contribute to the robustness of the scientific discourse.
Comments on the Quality of English LanguageMinor editing of English language required
Author Response
Reviewer Comment |
Response |
Reviewer 1
|
|
The authors have undertaken an investigation into the interaction of certain ligands with Estrogen Receptor-α36. It is imperative to underscore that the manuscript exclusively presents in silico data, devoid of any experimental results. It is my contention that, for the successful submission to a high-impact journal, the inclusion of experimental evidence is paramount to substantiate the novel insights under development. |
The authors appreciate the reviewers for taking their tight schedule to critique our work, and we confirm that the reviewer’s summary of our work is correct. However, I am unsure if reviewers decide whether a research work is publishable in a specific journal with low or high IF. To my understanding, reviewers can confirm the quality of a paper to the journal after reviewing a manuscript. Moreover, several in silico findings published in high-IF medicinal chemistry are available in the literature. For example, we have published purely in silico studies in a high-IF medicinal chemistry journal. Lastly, this study is submitted under a Special issue, “In Silico Drug Design and Discovery: Big Data for Small Molecule Design II”, indicating that the manuscript may be just in silico or combined in silico and wet lab-based findings. Nevertheless, we have an ongoing experimental wet lab-based study on the mechanism of BF-B inhibition of estrogen receptor alpha36 relative to its mutations, and we are expecting promising findings, which we envision will also be shared with the broader scientific community through a peer-reviewed publication. |
(1) It is noted that the structures depicted in Figure 1 are inverted in accordance with their respective legends. The rationale behind representing the chemical structures with all hydrogen atoms in a 2D model remains unclear. |
The authors appreciate the reviewer’s comments, and necessary corrections have been made in the main manuscript. |
(2) The legend accompanying Figure 2 is incomplete, merely stating, “Figure 2. Superimposition of….” A comprehensive and informative legend is essential for a complete understanding of the figure.
|
The authors thank the reviewer for his observation. The correction has been made in the manuscript. |
(3) The Principal Component Analysis (PCA) depicted in Figure 12 requires more thorough discussion and elucidation. The absence of PC1 and PC2 values on the x and y-axis of Figure 2 warrants attention. Additionally, the matrix variables utilized in the analysis are inadequately described.
|
The authors appreciate the reviewer for raising this comment. A discussion amendment has been made using “Figure 11”, indicating comparative visual structural elucidation of motion shifts in each system. In addition, the two matrix variables have been described. |
(4) It is strongly recommended that experimental evidence be incorporated into the results section to enhance the overall quality of the manuscript. This inclusion will fortify the credibility of the findings and contribute to the robustness of the scientific discourse.
|
The authors thank the reviewer for this suggestion. The works of Guo et al. and Jeong and Ryu have been referenced in the manuscript, providing literature on the experimental findings. However, we have ongoing wet lab studies on this work, which are meant to be published separately because the volume of the work would be too massive to combine with in-silico findings. |
Regards.
Authors
Reviewer 2 Report
Comments and Suggestions for Authors
In this manuscript, the authors reported a thorough comparison between the structures of
broussoflanol B (BFB)-bound estrogen receptor (ER) alpha-36 and 66. The information provided in this study might help the designing of potent structure-based compounds targeting. The specific comments are list below.
1. The abbreviations for estrogen receptor (ER) a-36 and 66 have to be consistent throughout the manuscript. Particularly, the position of the hyphen within the abbreviations should be consistent.
2. In line 47, Erb should be ERb. Also in line 66, Er-a36 should be corrected.
3. Other typos should also be corrected. Line 84, ER-α66.70,71. Line 88, ICI 182,780.6,73. Line 121, The three (3) selected templates. Line 157, minimized.
3. A lot of background information was provided between lines 49-57, but no reference was cited.
4. Within the abstract, the following sentence does not make sense: “The RMSF findings corroborate the RMSD results because the bound and unbound ER-α36 systems showed to be less stable than their ER-α36 counterparts.”
5. The discussion should be revised to better support the statement at the end of Abstract: “These findings present a model that describes the mechanisms by which the BFB compound induces downregulation-accompanied cell cycle arrest at the Gap0 and Gap1 phases.”
None
Author Response
Reviewer Comment |
Response |
Reviewer 2 |
|
In this manuscript, the authors reported a thorough comparison between the structures of broussoflanol B (BFB)-bound estrogen receptor (ER) alpha-36 and 66. The information provided in this study might help the designing of potent structure-based compounds targeting. The specific comments are listed below. |
The reviewer’s comments in this manuscript are correct. |
1. The abbreviations for estrogen receptor (ER) a-36 and 66 have to be consistent throughout the manuscript. Particularly, the position of the hyphen within the abbreviations should be consistent. |
The authors appreciate the reviewer’s critical comment and have made the corrections accordingly. |
2. In line 47, Erb should be ERb. Also in line 66, Er-a36 should be corrected. |
The authors appreciate the reviewer for pointing out the oversight and have corrected it. |
3. Other typos should also be corrected. Line 84, ER-α66.70,71. Line 88, ICI 182,780.6,73. |
The authors thank the reviewer for this comment; corrections have been made in the manuscript. |
4. A lot of background information was provided between lines 49-57, but no reference was cited.
|
The authors agree with the reviewer on this comment and have provided at least one article for each critical literature claim. |
5. Within the abstract, the following sentence does not make sense: “The RMSF findings corroborate the RMSD results because the bound and unbound ER-α36 systems showed to be less stable than their ER-α36 counterparts.” |
The authors appreciate the reviewer for this comment and have restructured the statement for clarity. |
6. The discussion should be revised to better support the statement at the end of Abstract: “These findings present a model that describes the mechanisms by which the BFB compound induces downregulation-accompanied cell cycle arrest at the Gap0 and Gap1phases.” |
The authors consider this comment essential and will improve the quality of our manuscript. Thus, we have justifiably discussed our findings in the main text in alignment with the results. |
Regards.
Authors